# A Longitudinal Study Examining the Association between Cognitive Behavior and Rational Abilities and the Effect of Sleep Quality on Construction Laborers

Sathvik Sharath Chandra [1,2], Krishnaraj Loganathan [2], Bankole Osita Awuzie [3] and Faming Wang [1,*]

1 College of Safety Science and Engineering, Xi'an University of Science and Technology, Xi'an 710054, China
2 Department of Civil Engineering, Faculty of Engineering and Technology, SRM Institute of Science and Technology, Kattankulathur 603203, Tamil Nadu, India
3 School of Construction Economics and Management, University of Witwatersrand, 10b Yale Rd, Braamfontein, Johannesburg 2000, South Africa
* Correspondence: dr.famingwang@gmail.com or faming.wang@xust.edu.cn

**Abstract:** Construction laborers are constantly subjected to irregular work hours, leading to insomnia and poor sleep quality, which impacts cognitive and rational behavior. This negatively influences decision-making capabilities, resulting in accidents on site. This study determined the effect of sleep quality on the cognitive behavior and rational ability of construction laborers. A quantitative research design comprised of a questionnaire survey was conducted for data collection purposes. Respondents comprised a randomly selected sample of construction workers, and a statistical analysis of the results was performed to investigate existing correlations. Data were collected using questionnaires from 575 and 310 respondents in the initial and latter phases, respectively, from five construction companies in Southern India, and analyzed using inferential statistics. Shift work negatively affects both the early and late phases of rational abilities. A negative correlation was observed between age and disturbed rationality in the late phase, despite not being observed in the early phase. Gender, rational ability, age, shift work, sleep quality, and cognitive behavior were not correlated in either the early or late phases. Furthermore, age, shift work, and sleep quality were not correlated with cognitive behavior. Rather, sleep quality and shift schedules were associated with rational ability and cognitive behavior impairment. There was a transient relationship between insufficient sleep and the ability to make rational decisions. This study contributes to the current discourse regarding the improvement of the sleep health of construction workers to enhance their well-being and productivity.

**Keywords:** construction laborers; rational abilities; cognitive behavior; sleep quality

## 1. Introduction

Sleep is essential to human health. A good night's sleep provides the body with the ability to recuperate from exhaustion, while aiding in the repair of body cells [1]. When a person sleeps for less than six hours every night, their logical functions, such as attention and memory, are negatively impacted [2]. In adults, sleeping less than seven hours per night adversely affects their daily functioning [3], as lack of sleep impairs not only reasoning and perception, but also self-control, empathy, and positivity [4]. In addition, sleep plays an important role in the control of individuals' cognitive abilities and their deliberate regulation of emotions [5]. Continual difficulties in initiating or maintaining sleep, poor sleep quality, and reduced sleep quality have been shown to lead to fatigue and to disrupt a persons' ability to work. Thus, the amount of work a person can accomplish depends on his or her sleep habits [6]. A recent study revealed that sleep deprivation impairs concentration, long-term memory, and workplace safety [7].

In the construction industry, previous studies have shown a significant correlation between sleep (quality and quantity) and the productivity of construction workers [6]. Construction laborers face numerous risks and dangers within the workplace, resulting in accidents,

which are often magnified by the scale of the construction site. Such accidents have detrimental effects on the well-being of employees and the local community, as well as the economic health of the employer, as frequent accidents increase employee absenteeism and decrease efficiency and profitability. Moreover, workplace hazards demotivate laborers [8], resulting in disorganization and lax execution of tasks [9]. Among all industrial sectors, construction is considered one of the most dangerous due to the high frequency of accidents associated with this sector [10]. In addition, various construction-related tasks remain hazardous, thereby presenting significant health risks to the construction laborer [11,12].

The increasingly hazardous nature of the construction project environment renders the possession of the adequate mix of cognitive behavior and rational abilities by workers imperative for accident prevention. Furthermore, it is essential that construction laborers possess optimal rational abilities because their work involves challenging assignments, rapid decision-making scenarios, and teamwork [13,14]. Laborers with sleep disorder symptoms rarely observe safety practices due to the impairment of their rational abilities [15]. Likewise, any decrease in the cognitive ability of laborers may negatively affect their performance at work, as well as their physical and mental health [16–18].

Sleep (quality and quantity) has been shown to impact the cognitive behavior and rational abilities of individual workers across different industrial sectors, including construction [16]. Construction laborers often work irregular hours, spread mostly across two shifts (day and night), and rest sporadically. Compared to day-shift laborers, night-shift laborers in the construction industry have a significantly greater incidence of sleep disorders, such as insomnia and poor sleep quality [19].

Accordingly, considerable attention has been paid to the relationship between sleep and cognitive behavior and rational abilities, specifically among construction laborers [20]. However, these studies have adopted a cross-sectional perspective as it relates to appraising the impact of sleep quality and quantity on the aforementioned facets. As such, a longitudinal investigation into the sustained effect of sleep quality on the cognitive behavior and rational ability of construction laborers remains lacking. This is the gap which this study seeks to fill.

Accordingly, the aim of the study was to examine the sustained impact of sleep quality on the rational abilities and cognitive behavior of construction laborers in the long term. The following objectives were considered toward achieving the aims of the study:

- Establish a patterns in the relationship between socio-economic characteristics of construction laborers and the extent to which sleep quality impacted their rational abilities and cognitive behavior.
- Examine the effects of sleep quality on work-related cognitive behavior and rational abilities of construction laborers in Southern India.
- Analyze how these influencing factors vary over time, that is, whether sleep is merely associated with transient state-like changes in cognitive behavior and rational abilities, or whether inadequate sleep predicts future declines in cognitive behavior and rational abilities at work.

This paper is divided into six sections. In Section 2, we examine the relevant literature relating sleep quality to the rational abilities and cognitive behavior of construction laborers. In Section 3, the research methodology used to collect and analyze the data is discussed. The results are presented in Section 4 and subsequently discussed in Section 5, alongside the implications. The limitations of the study are presented in Section 6. Finally, the conclusions are presented in Section 7.

## 2. Literature Review

Cognitive behavior has been described as the ability to perceive and react, process and understand, store and retrieve information, make decisions, and produce appropriate responses [21]. Insufficient sleep can culminate in behavioral habits that undermine effective decision making within work teams [16]. Reduced cognitive behavior at work because of poor sleep may constitute a serious hazard. Well-functioning emotional regulation is important

in the workplace because it imbues workers with collaborative, coping, creative, and work engagement capabilities [22]. Construction workers are expected to control their cognitive behavior, even when confronted with rationally challenging work [16]. However, sleep-deprived workers have been known to show a lower threshold for stressful events, while experiencing increased negative cognitive behavior, such as anger and anxiety, compared to those who are not sleep-deprived [23]. The potentially harmful effects of poor sleep quality represent a safety risk factor, as well as poor mental and physical health at work [24].

Rational characteristics refer to an individual's ability to change body behavior, and a decline in these characteristics results in an insufficient stress level, inability to concentrate, difficulty remembering, and significant memory loss. Rational abilities are gradually affected by lack of sleep, which negatively impacts work productivity. Sleep quality is essential for construction workers to maintain a healthy and well-rested condition and enhance their rational abilities and productivity at work [25]. The absence of rational abilities can also lead to stress, the inability to concentrate and remember, and an increase in the number of sick days resulting from inadequate sleep. Impaired rational abilities adversely affect the efficiency and productivity of laborers. Further, certain near-miss incidents, such as the misuse of personal protective equipment and the operation of ladders while drowsy, may be linked to poor sleep quality. According to Paavonen et al. [26], impaired rational abilities over prolonged periods are directly linked to the decline in workforce efficiency and productivity in the workplace. Researchers such as Manoharan et al. [27] and Chakraborty et al. [28] highlighted the importance of maintaining proper rational characteristics among laborers as related to sleep quality.

The productivity levels of construction laborers are adversely affected by the impaired rational abilities that adversely affect their behavior in the construction environment [29]. Previous studies related to the construction workplace have focused on (a) whether cognitive behavior and rational abilities have an impact on the productivity of construction laborers, (b) the effect of sleep quality on the rational abilities of laborers at work, and (c) factors related to rational abilities among Korean construction laborers [7,18]. None of these studies attempted to determine the prevalence of this phenomenon within the construction workplace, particularly in developing countries such as India, where labor-intensive practices are still prevalent [15]. However, studies have shown the cumulative impact of decreased rational abilities due to sleep deprivation on concentration and memory regarding task execution and safety [14,18].

Globally, there are health and safety regulations and standards that are specific to each country. The purpose of these regulations and standards is to protect the health of workers. Safety and health standards are set by the Occupational Safety and Health Administration (OSHA) in the United States [7]. Workplace safety and health information and guidance is provided by the European Agency for Safety and Health at Work [9]. In Australia, WorkSafe Australia is responsible for setting standards for workplace safety and health [10]. The Building and Other Construction Workers Act, 1996 regulates the safety, health, and wellbeing of construction workers in India [27].

The ability of construction laborers to make rational decisions is essential because their work focuses on challenging assignments, rapid decision-making situations, and group cooperation. Therefore, an individual's productivity and physical and mental health may be adversely affected by a decrease in cognitive behavior at work resulting from impaired rational abilities, thus posing a significant risk.

Although this risk persists in the construction industry, particularly in developing countries where labor-intensive construction remains prevalent, there is a lack of knowledge regarding how sleep quality affects cognitive behavior and rational abilities among construction laborers, resulting in impaired attention, lower alertness, and poor productivity and concentration among construction workers over sustained intervals [15].



## 3. Materials and Methods

A longitudinal study has been adopted for studying the relationship between sleep quality, cognitive behavior, and rational abilities of workers in two phases, known as the early phase and the late phase, within the construction environment [12].

The five southern states of India (Tamil Nadu, Telangana, Karnataka, Andhra Pradesh, and Kerala) were selected as the geographical context within which this study was situated. Initially, 25 government-owned construction companies operating in these states were selected using predetermined selection criteria. A significant component of these criteria was the corporation's inability to produce a high level of labor productivity and the accompanying poor financial productivity. Among the companies initially approached, five of these companies agreed to participate in the survey. There were 160 participants in each company who agreed to participate in the study. Construction laborers employed by these five corporations were surveyed using questionnaires, with an interval of nine months between surveys.

The initial data collection was conducted in October 2020, with the subsequent collection in May 2021. An informed consent form was provided to the participants. The sample size for this study was determined by conducting a power analysis, at a significance level of $\alpha = 0.05$ [14]. Consequently, 620 of the 800 construction laborers within the consenting organizations participated in the study. Participants were classified into skilled, unskilled, and semi-skilled laborers, based on their abilities and the nature of the work. The skilled laborer has specialized or advanced training to work on the construction site, the unskilled laborer does not possess specialized training, and the semi-skilled laborer has some degree of specialized training or the ability to work on the site [22]. The characterization of the laborers in the early and late phases of the data collection is listed in Table 1.

**Table 1.** Characterization of the participating laborers (respondents).

|  | Early Phase (n = 620) | Late Phase (n = 310) |
|---|---|---|
| Gender (%) | | |
| Male | 410 (66%) | 175 (56%) |
| Female | 210 (34%) | 135 (43%) |
| Age (y) | | |
| Mean (SD) | 43.90 (14.23) | 45.84 (13.71) |
| Max–min | 22–70 | 24–69 |
| Median | 44 | |
| Work schedule (%) | | |
| (Day/Evening) non-shift | 235 (37%) | 172 (56%) |
| (Day/Evening/Nights) shift | 385 (62%) | 138 (44%) |
| Time spent by construction laborers | | |
| Median (SD) | 13.85 (14.20) | 13.65 (19.45) |
| Labors current time in | 0–49 | 0–44 |
| Max–min Median Mean (SD) | 12 | 14 |
| Type of laborers (%) | | |
| Skilled | 325 (52%) | 165 (53%) |
| Unskilled | 208 (33%) | 105 (16%) |
| Semiskilled | 87 (14%) | 40 (6%) |

To assess the sleep quality, rational abilities, and cognitive behaviors of the laborers, a full-length questionnaire was developed. The information collected from the survey was standardized during the analysis. Therefore, the use of questionnaires was crucial to the

successful collection of data in this study. Surveys are an effective method to provide a snapshot of the circumstances at any given time by describing the phenomenon of live associations, evaluating them, and predicting their outcomes [30]. A Cronbach's Alpha reliability test, which was conducted to test for reliability of the data collection instrument, showed excellent internal consistency, posting a Cronbach's Alpha of 0.935. Cronbach $\alpha$ values greater than 0.7 indicate excellent data reliability [22]. This indicates a high reliability of 93.5% when it comes to the answers given by the respondents.

The descriptive data analysis only included 575 participants in the early phase and 310 in the late phase, respectively, as shown in Table 2, due to the fact that 3% and 19% of the collected data were missing from the early phase and late phase surveys, respectively.

**Table 2.** Description of sample collection in the early and late phases.

| Description | Early Phase | Late Phase |
|---|---|---|
| Distributed questionnaires | 620 | 620 |
| Returned valid questionnaires | 575 | 310 |
| Returned invalid questionnaires | 31 | 190 |
| Unreturned questionnaires | 14 | 120 |

### 3.1. Questionnaire Survey

The questionnaire was developed and distributed to the laborers to assess their sleep and psychological behavior. Survey questionnaires provide standardized knowledge that is used in survey analysis and are thus considered essential for data collection. A pilot study involving 65 construction labors was conducted to evaluate the feasibility of retaining the intended meaning of the questionnaires after translation into several native languages, including Kannada, Telugu, Tamil, Hindi, and Malayalam. Table 3 lists the survey parameters, and the questionnaire formats with references. The questionnaires were designed in both English and the vernacular, which was necessary owing to the different levels of formal education possessed by the construction laborers surveyed. Table 4 provide the research basis and format of the questionnaires.

**Table 3.** Parameters considered for questionnaire survey.

| Parameters Considered for Questionnaire Survey | References |
|---|---|
| Gender | [11,12] |
| Age | [21,23] |
| Type of laborers | [30,31] |
| Shift schedule | [18,20] |
| Sleep quality | [6,32] |
| Rational abilities—disturbed | [22,33] |
| Insomnia | [4,15] |
| Cognitive abilities—disturbed | [34,35] |

### 3.2. Parameters

The three basic parameters of the study were gender, age, and shift work pattern [28]. The gender and age of the subjects were selected as control variables due to their strong correlation with cognitive behavior and rational abilities [16,31]. Three groups of shift schedules were considered in this research: (1) day shifts alone; (2) shift schedules without night shifts; and (3) shift schedules with night shifts. Night shifts were not optional for 42% of the temporary laborers, 8% of dayshift laborers had the same working hours as the night-shift laborers, and 50% worked shifts alternating between day and night shifts.

**Table 4.** Questionnaire parameters and formats.

| Category | Survey Parameter | Format |
| --- | --- | --- |
| Demographic | Gender | Text free |
| | Age | Text free |
| Labor | Type of laborers | MCQ * |
| | Work schedule | MCQ |
| | Construction laborer time | MCQ |
| Sleep | Sleep quality | MCQ |
| | Insomnia | Insomnia scale |
| Psychological | Disturbed rational abilities | Likert scale |
| | Disturbed cognitive behavior | Likert scale |

* MCQ—multiple choice question.

### 3.3. Insomnia Scale

The Insomnia Scale [33,36–38] is a measure of insomnia symptoms included in the Diagnostic and Statistical Manual of Mental Disorders, Fifth Edition (DSM-5). Comprising a set of six questions, the Insomnia Scale is widely believed to be a dependable tool for assessing sleep quality [32,34,39–42]. The questionnaire sought to elicit information relating to the construction laborers' perspectives on their typical workweek attendance (0–7) and whether they had experienced these six explicit signs of sleep deprivation in the preceding month: waking up early, having trouble staying asleep, going to sleep late, being too tired, having insufficient sleep, and not getting enough rest [43]. The laborers were also asked to indicate the number of days per week (0–7) on which they had experienced these adverse effects. A continuous scale was used during the study, with computation scores ranging from 1 to 43. Both early and late phases of insomnia were associated with a high level of dependence ($\alpha = 0.85$).

### 3.4. Cognitive Behavior and Rational Abilities Related to Construction Work

A burnout assessment tool was used to estimate cognitive behavior and rational abilities [35,44]. The following statements were used to estimate disturbed cognitive behavior: "in the workplace, I have trouble staying engrossed", "in my workplace, I struggle to think clearly", "I am forgetful and distracted at work", "when I am working, I have concentration difficulties", and "I make mistakes in my work because I have my mind on other things". The following assertions were used to approximate the disturbed rational abilities: "at my workplace, I feel unable to control my emotions", "I do not comprehend the way I react emotionally at work", "at my workplace, I become irritable when things do not go my way", "I get disappointed or annoyed in the workplace without knowing why", and "at my workplace, I may overreact unintentionally". The laborers were tasked with assigning a rating to each item, using a five-point Likert scale, that varied from one (never) to five (consistently), based on how frequently they believed each remark to be applicable to the circumstances they were facing at work. The item-specific means of the scores were subsequently determined [24,33]. According to our measurements, in the early and late phases, the internal consistency of the rational abilities—disturbed was 0.89 and 0.91, respectively, whereas the disturbance of cognitive behavior was 0.82 and 0.80, respectively. Both outcomes were satisfactory.

### 3.5. Statistical Analysis

This study was conducted using IBM Statistical Package for Social Sciences Statistics for Windows Version 27.0, which was used for all analyses. Pearson's correlation analysis was performed to determine the relationship between the fundamental parameters used in the earlier and later phases of the experiment. A *t*-test was performed to determine

whether there was a difference between the construction laborers who participated in only the first phase of the project compared to those who participated in both phases. For the early and late phases, multiple hierarchical regression analyses were conducted on the disturbed rational abilities and cognitive behavior using two and three steps, respectively. Age, gender, shift patterns, and sleep quality were considered in the first step of the regression analyses, and sleep disorders were included in a separate step. Later in the analysis process, adjustments were made to account for early signs of disturbed rational abilities and cognitive behavior, according to estimates made at the beginning of the study.

The late phase of the analysis included adjustments for the early symptoms of disturbed rational abilities and cognitive behavior, as estimated at the start of the longitudinal study.

$$r = \frac{\sum (x_i - x)(y_i - y)}{\sqrt{\sum (x_i - x)^2 \ \sum (y_i - y)^2}} \tag{1}$$

where $r$ = correlation coefficient, $x_i$ = $x$-variable values in a sample, $x$ = mean values of the $x$ variable, $y_i$ = $y$-variable values in a sample, and $y$ = mean values of the $y$ variable.

## 4. Results

Both males and females were grouped according to their age and rotational shift work schedules. Table 5 lists the interpretive insights and relationships among factors that were examined in the early and late phases of the study. No significant difference was found between laborers who completed both the early and late questionnaires in terms of their cognitive behavior and rational abilities compared with those who only completed the early questionnaire. The difference in age between laborers involved in both surveys was negligible (mean [M] = 44.90 years, standard deviation [SD] = 14.23) compared to those who completed only the early questionnaire (M = 45.84 years, SD = 13.71), as listed in Table 1. There was a negative correlation between shift schedules and rational abilities in both early and late phases. The results of these analyses for work-related rational abilities and cognitive behavior are presented in Tables 6 and 7, respectively.

**Table 5.** Descriptive statistics and statistical correlations for measured variables during the early and late phases.

| | 1 | 2 | 3 | 4 | 5 | 6 | 7 | 8 | 9 | 10 | 11 |
|---|---|---|---|---|---|---|---|---|---|---|---|
| Gender [a] | | | | | | | | | | | |
| Age | 0.04 ** | | | | | | | | | | |
| Shift schedule [b] | 0.03 ** | | | | | | | | | | |
| Sleep quality [b] | 0.14 | | | | | | | | | | |
| Sleep quality [c] | 0.18 | 0.02 ** | 0.09 | 0.7 * | | | | | | | |
| Insomnia [b] | 0.14 | 0.03 ** | 0.04 ** | 0.39 | 0.29 | | | | | | |
| Insomnia [c] | 0.18 | 0.05 ** | 0.03 ** | 0.44 | 0.39 | 0.75 | | | | | |
| Rational abilities—disturbed [b] | 0.14 * | 0.09 * | 0.25 | 0.15 * | 0.14 * | 0.29 | 0.20 | | | | |
| Rational abilities—disturbed [c] | 0.15 * | 0.12 | 0.19 | 0.14 * | 0.14 * | 0.38 | 0 | 0.56 * | | | |
| Cognitive abilities—disturbed [b] | 0.16 * | 0.02 ** | 0.08* | 0.08 * | 0.18 * | 0.29 | 0.18 * | 0.58 | 0.36 | | |
| Cognitive abilities—disturbed [c] | 0.3 ** | 0.03 ** | 0.13 * | 0.16 * | 0.23 | 0.25 | 0.22 | 0 | 0.49 | 0.63 * | |
| Mean (SD) | 0.49 (0.52) | 10.9 (11.4) | 0.52 (0.52) | 6.58 (0.84) | 6.55 (0.86) | 11.65 (8.69) | 10.65 (8.53) | 1.77 (0.64) | 2.09 (0.61) | 1.48 (0.49) | 1.59 (0.52) |

\* $p < 0.05$; ** $p < 0.01$; [a] 0 male, 0 female; [b] 0 early study; [c] 0 late study.

**Table 6.** Relationships between the variables measured in the early and late phases relating to disturbed rational abilities work at the site (determined through regression analysis).

| | Rational Abilities—Disturbed | | | | | | | |
| | Early Phase (N = 409) | | | | Late Phase (N = 215) | | | |
| | B | SE | *p* | $R^2$ | B | SE | *p* | $R^2$ |
|---|---|---|---|---|---|---|---|---|
| **1-Model** | | | 0.62 | | | | 0.08 | |
| Gender | 0.09 | 0.08 | 0.14 | | 0.12 | 0.10 | 0.16 | |
| Age | 0.04 | 0.04 | 0.65 | | 0.22 | 0.05 | 0.10 | |
| Shift schedule | 0.23 | 0.08 | 0 | | 0.27 | 0.2 | 0.02 | |
| **2-Model** | | | 0.15 | | | | 0.26 | |
| Gender | 0.04 | 0.07 | 0.51 | | 0.03 | 0.09 | 0.80 | |
| Age | 0.03 | 0.04 | 0.75 | | 0.3 | 0.05 | 0.08 | |
| Shift schedule | 0.23 | 0.08 | 0 | | 0.25 | 0.11 | 0.02 | |
| Insomnia | 0.28 | 0.05 | 0 | | 0.42 | 0.06 | 0 | |
| Sleep quality | 0 | 0.05 | 0.98 | | 0.08 | 0.07 | 0.39 | |
| **3-Model** | | | | | | | 0.43 | |
| Gender | | | | | 0.03 | 0.08 | 0.86 | |
| Age | | | | | 0.15 | 0.05 | 0.04 | |
| Shift schedule | | | | | 0.12 | 0.09 | 0.2 | |
| Insomnia | | | | | 0.32 | 0.05 | 0 | |
| Sleep quality | | | | | 0.2 | 0.06 | 0.13 | |
| Cognitive abilities—disturbed [b] | | | | | 0.48 | 0.07 | 0 | |

[b] 0 early study.

**Table 7.** Relationships between the variables measured in the early and late phases relating to disturbed cognitive behavior at the site (determined through regression analysis).

| | Cognitive Behavior—Disturbed | | | | | | | |
| | Early Phase (N = 409) | | | | Late Phase (N = 215) | | | |
| | B | SE | *p* | $R^2$ | B | SE | *p* | $R^2$ |
|---|---|---|---|---|---|---|---|---|
| **1-Model** | | | 0.06 | | | | 0.08 | |
| Gender | 0.14 | 0.06 | 0.03 | | 0.19 | 0.09 | 0.16 | |
| Age | 0.06 | 0.04 | 0.44 | | 0.08 | 0.05 | 0.43 | |
| Shift schedule | 0.09 | 0.07 | 0.18 | | 0.15 | 0.09 | 0.14 | |
| **2-Model** | | | 0.13 | | | | 0.8 | |
| Gender | 0.1 | 0.06 | 0.08 | | 0.14 | 0.09 | 0.10 | |
| Age | 0.05 | 0.03 | 0.50 | | 0.07 | 0.05 | 0.48 | |
| Shift schedule | 0.09 | 0.07 | 0.19 | | 0.13 | 0.09 | 0.15 | |
| Insomnia | 0.29 | 0.04 | 0 | | 0.20 | 0.06 | 0.03 | |
| Sleep quality | 0.05 | 0.04 | 0.44 | | 0.04 | 0.06 | 0.72 | |
| **3-Model** | | | | | | | 0.42 | |
| Gender | | | | | 0.03 | 0.08 | 0.82 | |
| Age | | | | | 0.06 | 0.05 | 0.50 | |
| Shift schedule | | | | | 0.07 | 0.09 | 0.39 | |
| Insomnia | | | | | 0.09 | 0.05 | 0.28 | |
| Sleep quality | | | | | 0.04 | 0.05 | 0.8 | |
| Cognitive abilities—disturbed [b] | | | | | 0.7 | 0.07 | 0 | |

[b] 0 early study.

### 4.1. Work-Related Rational Abilities

An analysis of multiple hierarchical regression, accounting for age, gender, shift schedule, sleep disorder, and sleep quality, explains the 15% variation in rational abilities—disturbed in the early phase ($F_{(7, 402)} = 13.44$) as well as the 26% variation in the unsettled rational abilities in the late phase ($F_{(7, 170)} = 12.30$). The variance increased to 43% when the disturbance in rational abilities was taken into account ($F_{(9, 192)} = 33.26$). Sleep disturbances measured in the early phase were positively correlated with impaired rational abilities recorded in both phases. No significant correlation was found between sleep quality on working days and disturbed rational abilities in either phase. However, there was a negative correlation between shift schedule and disturbed rational abilities in both phases. There was an inverse relationship between the age of the person and the disturbed rational abilities, but only in the late phase. The correlation between gender and rational abilities was not significant in either phase.

### 4.2. Work-Related Cognitive Behavior: Sleep Quality and Alternating-Shift Schedules

Sleep disorders were included in the model, explaining the 13% variance in disturbed cognition in the early stage, and the 8% variance in unsettled cognition in the later stage ($F_{(8, 180)} = 3.71$). The variance increased to 42% after accounting for the early phases of cognitive behavior in the late model ($F_{(8, 180)} = 16.40$). The correlation between sleep disorders measured early in the day, as well as disturbed cognitive behavior, was significant. Disturbed cognitive behavior did not differ significantly between the early and late phases. Disturbances in sleep quality, early symptoms of disturbed cognitive behavior, and sleep disturbances were negatively correlated with disturbed cognitive behavior before adjusting these factors in both phases. Neither the early nor the late phase appeared to be influenced by age, shift work, sleep quality, or cognitive behavior.

An analysis of cross-lagged panel data did not reveal significant differences in the disturbed rational abilities ($\beta = -0.03$, $p = 0.834$) or cognitive behavior ($\beta = -0.05$, $p = 0.606$) in the late or early phases. A disturbance in the presence of rational abilities ($\beta = -0.03$, $p = 0.866$) and cognitive behavior ($\beta = -0.06$, $p = 0.648$) did not predict sleep quality nine months after the early phase.

Although the participants reported relatively poor health statuses, 92% rated themselves in "good" or "very good", health, with 8% reporting "bad" health.

## 5. Discussion

This study examined the cognitive behavior of construction laborers in relation to their rational abilities and the sustained effects of sleep quality on this relationship. Relationships between sleep quality and cognitive behavior and between sleep quality and rational abilities of construction laborers were established separately. Regression analysis was performed to predict the mean values of the independent and dependent variables by identifying their relationship. Despite adjusting for the scores obtained during the early phase, sleep quality was associated with impaired cognition behavior and rational abilities related to work productivity. Nevertheless, the latter correlation continued to be significant after nine months. Subjective sleep quality, shift schedules, and cognitive behavior or rational abilities were not correlated in the late phase. The findings substantiate the direct correlation between sleep, rational abilities, and cognitive behavior observed after nine months. The relationship between insomnia and cognitive behavior was also found to be more transient than that between insomnia and rational abilities.

Strong correlations between insomnia and impaired rational and psychological performance on construction sites have been identified in previous crossover and termination studies [37]. The association between insomnia and the deterioration of cognitive traits at work is no longer significant after correcting for early symptoms of cognitive disturbance [38,39]. There was a subtle improvement in the relationship between insomnia and impeded rationality after eight months [34]. When faced with difficult circumstances and important responsibilities, such as driving or operating large machinery, chronic sleep de-

privation has been shown to adversely affect performance. Sleep disturbance increased the risk of performance degradation and adversely impacted the safety of construction laborers. Depriving laborers of sleep impaired their executive abilities and prevented them from concentrating and making sound decisions, which could have fatal consequences [45–49]. It is imperative that construction laborers control their emotions and act rationally. According to our findings, insomnia does not appear to be causally related to the disruption of cognitive behavior; however, our results reveal that emotional regulation is a fundamental component of social relationships, mental health, and general health and well-being [25]. Poor sleep quality and sleep deprivation have been associated with rational disability and cognitive distress. This study sheds new light on the relationship between sleep and rational abilities, as well as cognitive behavior in work environments.

An association between sleep quality and gender its effect on cognitive behavior in the workplace in both the early and late phases of the study was found. The results indicate that the laborers had a disturbed sleep quality. In some cases, sleep quality may be more important than a longer duration of sleep because optimal sleep quality varies according to each individual's needs. Moreover, organizations should ensure that their employees receive adequate rest during shifts [40,46]. Sleep disorders are less likely to occur when laborers improve the quality and quantity of their sleep.

This study has many merits, including the use of a longitudinal design that enabled the examination of the relationship between sleep and rational abilities in a new way, as well as the cognitive behavior associated with sleep and work over the course of two time periods [50,51]. The observation of longitudinal changes between measurements may be deemed sufficient in this context [52,53]. However, it is recommended that longer intervals between measurement points be used in future studies. Linear multiple regression analyses provide more information than simple regression analyses regarding the relationship between insomnia and logical functioning. The variables included in the study did not show significant differences, except for age differences, despite a 50% dropout rate between the early and late phases [54,55]. Numerous rational abilities and cognitive disabilities were measured in this study in accordance with the construction workplace; however, the findings may also be applicable for improving work efficiency and safety at construction sites.

The classification of workers as skilled, unskilled, and semi-skilled is a complex issue, affecting labor markets, employment policies, and social and economic outcomes. These implications need to be considered by policymakers when designing and implementing policies aimed at promoting economic growth. Furthermore, policies designed to improve the skills and training of workers can have different effects, depending on their levels of education and skill.

## 6. Limitations

The self-reported sleep quality information used in this study may have resulted in inaccurate estimates of sleep quality. Participants in sleep studies overestimated their sleep quality in comparison to quality results measured objectively, which may have resulted in a bias in the results due to the use of self-reporting questionnaires. Meanwhile, questionnaires are advantageous because they provide specific information regarding the respondents. A study with a larger sample size was conducted, but a smaller sample size was advised. Several factors may have contributed to these results, including trauma, past accidents, physical illnesses, and psychosocial factors. Sleep problems and depression can be caused by traumatic brain injuries and anxiety. Construction laborers, however, would be disqualified from performing most of their tasks if they suffered from severe illnesses or trauma. Participants who were absent from work for an extended period due to illness were excluded from the study.

In addition, the data were collected prior to the COVID-19 pandemic. Therefore, the current dataset can only serve as a baseline for assessing the relationship between sleep quality and construction laborers' teamwork quality prior to the COVID-19 pandemic era.

It remains to be seen how the COVID-19 pandemic has changed the quality of sleep among construction workers.

### 7. Conclusions

This study concluded that sleep quality had a greater impact on rational abilities and cognitive behavior than did work schedules. Sleep deprivation and restlessness are associated with several cognitive characteristics; however, these characteristics were primarily evident in cross-sectional surveys. The longitudinal study revealed a stronger correlation between insomnia and rational abilities than the cross-sectional study, suggesting that sleep problems may have adverse effects over time. These results indicate that shift work negatively affects both the early and late phases of rational abilities. In particular, the disturbed rational trait in the early phase accounted for 13% of the total variance ($F_{(6, 402)} = 13.44$) and 26% of the variance in the unsettled rational abilities in the late phase ($F_{(7, 170)} = 12.30$). This was determined through multiple hierarchical regression.

A negative correlation was observed between age and disturbed rationality in the late phase, despite not being observed in the early phase. There was no correlation between gender and rational abilities in either phase. There was a 13% variance in the rational abilities in the early phase of the analysis, and an 8% variance in the disturbed rational abilities in the late phase of the analysis ($F_{(8, 180)} = 3.71$). Age, shift work, sleep quality, and cognitive behavior were not correlated in either the early or late phases. Considering that sleep deprivation adversely affects the cognitive behavior and rational abilities of construction laborers, prevention and treatment of insomnia should be prioritized.

There is a need for future research to examine the cumulative effects of inadequate and poor-quality sleep on rational abilities. Companies owe it to their employees to raise awareness of the sleep deprivation epidemic and to provide strategies for reducing the associated risks; these measures guarantee that laborers receive adequate sleep, thereby enhancing their productivity. Furthermore, the use of mobile devices, physical activity levels, and types of professions can be examined in relation to sleep quality. Finally, this study provides baseline data that can be used for comparison in the future to assess the impact of the COVID-19 pandemic on sleep quality.

**Author Contributions:** Conceptualization, S.S.C. and K.L.; methodology, K.L.; software, S.S.C. and B.O.A.; validation, S.S.C., K.L. and F.W.; formal analysis, B.O.A.; investigation, K.L.; resources, B.O.A. and F.W.; data curation, B.O.A.; writing—original draft preparation, S.S.C., K.L. and F.W.; writing—review and editing, B.O.A.; visualization, S.S.C.; supervision, K.L.; project administration, K.L. All authors have read and agreed to the published version of the manuscript.

**Funding:** This research received no external funding.

**Institutional Review Board Statement:** This research was approved by the Ethics Committee of the SRM Hospital and Research Center (2186/IEC/2020) and was conducted according to the principles of the Institutional Ethical Committee. Informed consent was obtained from all the participants for the study.

**Informed Consent Statement:** Informed consent was obtained from all the participants involved in the study.

**Data Availability Statement:** The data are available on request.

**Conflicts of Interest:** The authors declare no conflict of interest.

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
