# Peer review of "A Longitudinal Study Examining the Association between Cognitive Behavior and Rational Abilities and the Effect of Sleep Quality on Construction Laborers"

_sustainability, doi:10.3390/su15076257_

Round 1

Reviewer 1 Report

Thanks for the interesting topic and the comprehensive study. However, below are some comments to further improve the presentation of the study in this paper, 

Pg 1 line 34 - Any reference to the second sentence?

Pg 2 line 83 onwards - are these objectives or research questions? State clearly the aim of the study. 

Pg 4 line 185 onwards - this paragraph can be moved after Table 1 as it's about Table 2. It is recommended to introduce a Table/Figure immediately before it appears. Also, this helps insert text in between two tables. 

Pg 9 line 324 - Apart from the survey, is there any evidence of increased risk and impacts caused to the safety of labours? 

Are there any country-specific health and safety standards and/or regulations available? Authors may mention these in the literature review as relevant.

Construction accidents can occur due to a range of reasons. In this longitudinal study, is there any evidence that risks are increased due to sleep disorders?

Any analysis and implications based on the categorization of labours in terms of skilled and unskilled? 

The reasons for selecting a longitudinal study and its implications based on the results are not clear. Is this to verify and prove the sleep deprivation and its impacts or are there any interventions between the study periods?

It is clear that a particular area of India is selected. However, is there any language diversity and which language was used in the surveys? Are there any impacts due to their literacy levels specifically with unskilled labours? How did you maintain the reliability of responses? 

Discussion can be further improved by comparing and contrasting with previous studies. 

Author Response

See attached the file.

Reviewer 2 Report

This manuscript explores the relationship between cognitive behavior and the rational abilities of construction laborers. The authors collected questionnaires from construction companies in southern India and analyzed the results with statistical models. The analysis discovered a transient relationship between insufficient sleep and the ability to make rational decisions. 

The authors have reviewed the literature in a well-organized manner. The research question of understanding the relationships between sleep quality and cognitive behavior looks significant. The authors collected data and conducted statistical analysis to understand the relationships.

The questionnaires were eventually collected from five corporations. The authors might want to discuss whether the samples are representative briefly.

Table 4 is very challenging to read. It isn't easy to understand what each integer in the first column represents. Why is the table incomplete, missing the correlation between age and shift schedule, for example? According to the table, none of the values are significant because no stars are reported in the table. Is it true? 

I also need help understanding Tables 5 and 6. What are the differences between these tables? Are the values reported in the rows of 1-Model, 2-Model, and 3-Model the r-squared values? Why are r-squared for the early and late phases differ so much with the same model? Why does the r-squared decrease as the number of variables increases? The author also might want to make it more obvious that the 3-Model makes use of the outcome variables in the Early phase, and, therefore 3-Model for the Early phase does not make sense.

Reviewer 3 Report

This research investigated the impacts of sleep quality on cognitive and rational abilities of construction workers. The research is interesting and valuable. There are some points to improve the paper quality.

First, the title is “the mediating effect sleep quality on the relationship between cognitive behavior and rational abilities”. This is problematic, because there is no hypothetical model about the mediating effect in this paper. Furthermore, the mediating effect was not examined properly.

Second, more information should be given about the late survey in Section of 3. Materials and Methods.

Three, there were some inconsistent results between early and late study, detailed explanation should be presented.  

Round 2

Reviewer 2 Report

I'm unsure if the integer numbers in Table 4 are easily understandable for readers. Even if they are, the table can be challenging to read.

I don't understand the correlations between shift schedule, sleep quality, and age presented in Table 4.

Could you please adjust the formatting of the R-squared value in the 3-Model of Table 5?

I'm confused about why the R-squared value of the 3-Model in Table 6 is lower than that of the 2-Model.

Author Response

See attached the file.

Reviewer 3 Report

Most of my comments have been revised. However, the authors are still suggested to further modify the title to make it consistent with the research content. In addition, the number of respondents that participated in initial and latter questionnaires should be 575 and 310 in the Abstract, please update it.  

Author Response

See Attached file, thanks.
